# Cancer Stem Cell Biomarkers Predictive of Radiotherapy Response in Rectal Cancer: A Systematic Review

**DOI:** 10.3390/genes12101502

**Published:** 2021-09-25

**Authors:** Marzia Mare, Lorenzo Colarossi, Veronica Veschi, Alice Turdo, Dario Giuffrida, Lorenzo Memeo, Giorgio Stassi, Cristina Colarossi

**Affiliations:** 1Medical Oncology Unit, Mediterranean Institute of Oncology, 95029 Viagrande, Italy; marzia.mare@grupposamed.com (M.M.); dario.giuffrida@grupposamed.com (D.G.); 2Department of Biomedical, Dental, Morphological and Functional Imaging Sciences, University of Messina, 98122 Messina, Italy; 3Pathology Unit, Mediterranean Institute of Oncology, 95029 Viagrande, Italy; lorenzo.colarossi@grupposamed.com (L.C.); lorenzo.memeo@grupposamed.com (L.M.); cristina.colarossi@grupposamed.com (C.C.); 4Department of Surgical Oncological and Stomatological Sciences (DICHIRONS), University of Palermo, 90127 Palermo, Italy; veronica.veschi@unipa.it; 5Department of Health Promotion, Mother and Child Care, Internal Medicine and Medical Specialties (PROMISE), University of Palermo, 90127 Palermo, Italy; alice.turdo@unipa.it

**Keywords:** radiosensitivity, cancer stem cells, organoids, rectal cancer (RC), neo-adjuvant radiotherapy, in vitro radiotherapy

## Abstract

Background: Rectal cancer (RC) is one of the most commonly diagnosed and particularly challenging tumours to treat due to its location in the pelvis and close proximity to critical genitourinary organs. Radiotherapy (RT) is recognised as a key component of therapeutic strategy to treat RC, promoting the downsizing and downstaging of large RCs in neoadjuvant settings, although its therapeutic effect is limited due to radioresistance. Evidence from experimental and clinical studies indicates that the likelihood of achieving local tumour control by RT depends on the complete eradication of cancer stem cells (CSC), a minority subset of tumour cells with stemness properties. Methods: A systematic literature review was conducted by querying two scientific databases (Pubmed and Scopus). The search was restricted to papers published from 2009 to 2021. Results: After assessing the quality and the risk of bias, a total of 11 studies were selected as they mainly focused on biomarkers predictive of RT-response in CSCs isolated from patients affected by RC. Specifically these studies showed that elevated levels of CD133, CD44, ALDH1, Lgr5 and G9a are associated with RT-resistance and poor prognosis. Conclusions: This review aimed to provide an overview of the current scenario of in vitro and in vivo studies evaluating the biomarkers predictive of RT-response in CSCs derived from RC patients.

## 1. Introduction

Colorectal cancer (CRC) is the third most commonly diagnosed malignancy and the second leading cause of cancer death worldwide [1]. One-third of CRCs occur in the rectum, with an incidence of approximately 35% and a median age at diagnosis of 70 years, despite predictions suggesting that this picture will rise in the future [2]. In 60% of cases, RC presents in a locally advanced form with neoplastic infiltration beyond the muscularis of the rectum and/or lymph node involvement (cT3-T4 and/or cN+) [3]. Due to tumour location in the pelvis and close proximity to critical genitourinary organs, treatment of locally advanced rectal cancer (LARC) is particularly challenging [4,5]. The globally recognized therapeutic strategy for treating LARC consists of neoadjuvant chemo-radiotherapy (CRT) or radiotherapy alone followed by mesorectal excision (TME) [2,3,4,5]. Such a preoperative treatment regimen has the advantage of low toxicity, high sphincter preservation rate and low locoregional recurrence rate. Radiotherapy (RT) has been reported to significantly promote the downsizing and downstaging of large RCs in neoadjuvant settings; however, it is not free from adverse effects (such as serious anorectal and genitourinary complications), which may negatively impact on patient’s quality of life [6]. Moreover, a limited number of patients respond positively to a CRT, with a 30–40% rate of relapse [7]. Many biological mechanisms causing radio-resistance have been identified so far. Resistance to radiation therapy is associated with alterations within the tumour and in the surrounding microenvironment, such as DNA repair, growth signalling pathways, inflammation, angiogenesis and oxygen tension [8]. Thus, in order to avoid unnecessary treatments, costs and adverse events, a major effort has been directed either toward the development of pharmacological agents able to enhance response to radiation, or to the identification of predictors of the response to CRT [9]. It is now widely accepted that tumour maintenance is due to cancer stem cells (CSCs), a minority subset of tumour cells with stemness properties, able to undergo self-renewal and multi-lineage differentiation, sustain malignant growth and mediate CRT response [10,11,12]. CSCs have been demonstrated to display a higher capacity of DNA damage repair and a robust capability to initiate local and metastatic spread. CSC elimination can reduce the incidence of recurrence, metastasis and chemoresistant phenotype, which in turn effectively enhances sensitivity to therapy in advanced CRC patients [9,13,14]. Cellular resistance to CRT remains the primary barrier to overcome in order to achieve treatment success and improve RC prognosis; an increasing body of evidence from both experimental and clinical studies indicates that the likelihood of achieving local tumour control by CRT depends on the complete eradication of CSC populations [15]. Preoperative CRT followed by surgery has become the standard treatment for LARC. Therefore, to avoid unnecessary treatments, adverse events and costs, there is an increasing and urgent need to find CSC-related biomarkers predictive of response to radiotherapy [7,9,15,16]. Unfortunately, despite the potential role of CSCs in resistance to preoperative CRT, only a few studies have reported the prognostic effect of putative CSCs in RC treated with preoperative CRT [17].

Putative stem cells identified by surface markers have been found in numerous studies to be more radioresistant than non-stem cells of the same cell line in in vitro colony assays [18,19]. In addition, several studies have shown a significant increase in the fraction of marker-positive cells after irradiation in vitro or in vivo, which would be in line with selection of radioresistant CSC while the more sensitive non-stem cells are more effectively killed [18,19,20]. Some of these studies have also shown a lower number of DNA double-strand breaks after irradiation, again supporting a higher radioresistance of CSC [20]. These findings indicated that CSCs, also called tumor-initiating cells, possess two fundamental properties that make them different from other tumor cells: they have unlimited capacity to self-renew and differentiate to all cell populations present in the original tumors. These properties make these cells a root of tumor growth and recurrence and, thus, an important marker for tumour diagnosis, prognosis, and treatment, as well as critical targets for cancer therapy. Strong evidence is emerging to support the dynamic nature of tumour stemness, which can be influenced by genetic and epigenetic alterations and by tumoral microenvironment [10]. Although tumour cell heterogeneity displays a high level of complexity, CSC populations remain critical targets and biomarkers for cancer treatments. Due to the detection of cancer stem cells via the expression of biomarkers, tumour initiating cells have been identified in leukemia (CD34^+^, CD38^−^) [21] and in solid tumours including glioma (CD133^+^, ALDH1^+^) [22], breast (EpCAM^+^/CD44^high^/CD24^low^, ALDH1^+^) [23], and head and neck squamous carcinoma (CD44+) [24]. Various markers have been used to identify intestinal cancer stem cells based in the main on the utilization of mouse models. Human colorectal stem cells were first isolated on the basis of CD133 expression, also known as the prominin-1 glycoprotein [25]. The selected cells expressing the CD133 marker were isolated from primary colon cancer samples, and were capable of growing as spheres and forming tumours once inoculated in mice, remaining undifferentiated when cultured in serum-free media. Moreover, Todaro et al. showed that CD133+ cells are capable of producing IL-4 in order to evade apoptosis. The treatment of CD133+ cells with a α-IL-4 neutralising antibody significantly enhanced the sensitivity to chemotherapeutic drugs and increased their antitumour efficacy [26]. Furthermore, CD44 has also emerged as an important marker for CRC. It is a cell surface glycoprotein that functions in cell–cell interactions, adhesion of the cytoskeleton to the extracellular matrix and cell migration. Transcription of CD44 is at least in part activated by β-catenin Wnt signalling, and its overexpression is an early event in the transformation of colorectal adenoma to carcinoma. Knockdown of CD44 prevents tumourigenesis and clonal formation. Furthermore, injection of only 100 CD44+ cells is sufficient to initiate tumour formation in nude mice, and single CD44+ cells form tumour spheres with stem cell characteristics, which develop into tumours once inoculated into nude mice [27]. Unfortunately, several of these markers are not specifically expressed only in the stem or cancer stem cell population. Indeed some biomarkers, while overlaying stem cell populations, also mark other non-stem cells, and several normal intestinal stem cell markers also mark CSCs. Lgr5+ cells have been shown to be representative of the cell of origin of intestinal tumourigenesis and have tumour-initiating potential [28]. The degree of expression of this protein appears to be related to disease recurrence after treatment with curative intent in CRC [29].

## 2. Materials and Methods

### 2.1. Study Design

This is a systematic review of the most recent studies focusing on the characteristics of CSCs and their response to radiotherapy. It follows the Preferred Reporting Items for Systematic Reviews (PRISMA) guidelines.

### 2.2. Selection and Exclusion Criteria

Studies from 2009 to 2021 investigating the radiosensitivity/radioresistance of cancer stem cells from rectal or colorectal cancer were selected. The search was limited to original articles written in English language and based on human studies including at least more than four patient-derived samples. Meta-analysis and review papers were excluded.

### 2.3. Search Strategy

A systematic literature search of the PubMed and Scopus databases was undertaken. Significant references were checked and included. Relevant published studies were searched from each databases’ inception since January 2009 to July 2021. Search strings were combined by using the basic Boolean operators (“AND”, “OR”). The database search was performed using the following search terms: (“rectal cancer” OR “colorectal cancer”) AND (“radiotherapy” OR “radiosensitivity”) OR (“stem” OR “organoids”) in Title and Abstract.

### 2.4. Quality Assessment and Risk of Bias

To assess the quality of the selected articles, we formulated 10 questions (Table 1).

For each question a positive answer was scored as 1 and a negative as 0 (Appendix A). Each author classified and rated each record independently from each other. All the authors discussed the differences observed in the scores in order to identify a final consensus score. The study selected achieved >80% of the total score (positive response to at least 8 questions out of 10; Table 1). The risk of bias was assessed by Robvis R package [30]. Studies were evaluated based on their relative low, moderate, or high risk of bias. 

## 3. Results

### 3.1. Selection, Bias and Quality of Articles

We retrieved 464 records in total, 48 records from Pubmed and 416 from Scopus by using the searching strategy reported in the Material and Methods section. After duplicate removal, 112 studies met the inclusion criteria. Each author independently and accurately evaluated the full text of the selected studies. Only the studies showing results significant to the field, with a quality assessment score > 80, and with a low or moderate risk of bias, were eligible (Figure 1, Figure 2 and Figure 3 and Appendix A). Eleven studies on cellular radiosensitivity, identifying several cell-surface biomarkers of CSCs as predictors of radioresistance in RC, were included in this systematic review. Out of these, 5 studies were about CD133, 4 about CD44 including 1 also focusing on ALDH1, 1 on G9a and 2 on Lgr5. Of these, 2 studies investigated how mechanisms of DNA repair are involved in the RT response (Figure 1 and Table 2).

### 3.2. Biomarkers Predictive of Radiotherapy Response in CSCs Isolated from RC Patients

Radiation therapy represents an essential tool in the current treatment modality for CRC patients. Research has revealed the presence of CSC populations in different tumours, including RC, which are responsible for disease relapse, poorer survival rate and therapeutic resistance to conventional chemo- and radiotherapies [31,32]. In 1959, Hewitt and Wilson developed a quantitative method for tumour cell transplantation, which correlated the median transplantation dose (TD50) with irradiation dose, by employing the serial dilution in vivo assay. Such a method was then successfully adopted to correlate TD50 and radiotherapy as cancer treatment in multiple tumour models [15,33]. These studies laid the foundation for the subsequent research on CSCs in radiation oncology, suggesting the importance of the number of tumour-initiating cells as prognostic factors for tumour radiocurability [15,34,35,36,37,38]. The detection of biomarkers that characterize CSCs may play a crucial role in predicting the clinical outcome of radiotherapy-treated patients (Table 2) [39]. Currently, CD133 represents the main biomarker for the identification of putative CSCs in various types of cancer, including RC [39,40]. As CD133 is associated with poor clinical outcomes, evaluation of the sensitivity to radiation therapy in RC may serve to predict a possible complete or partial tumour response to RT, thus avoiding unnecessary treatments [20,31,40]. In 2010, Chen et al. explored the causes of radioresistance in human CRC by observing the changes in chromatin histone in human colorectal CSCs (CD133+) and non-CSCs (CD133−) after a single high-dose of radiation. A distinct difference was found in colorectal CSC chromatin structure, with compact patches in the CD133+ nucleus and loosely latticed structure in the CD133− nucleus. Such a mechanism seems to be related to heterochromatin formation and histone methylation, thus demonstrating that CSCs play a role in CRC radiosensitivity [41]. In the same year, tumour cells from 50 RC patients who had undergone preoperative chemoradiotherapy followed by surgery were analysed by Saigusa et al. CD133 expression, both on the luminal surface and in the cytoplasm, was found to be associated with a poorer response [42]. Furthermore, in 2012, Saigusa and colleagues demonstrated that gene expression levels of LGR5 in cancer cells and of CD44 in stromal cells were significantly correlated with disease recurrence, whereas only elevated levels of stromal CD44 was an independent prognostic factor of recurrence and overall survival of RC patients after preoperative CRT [43]. A retrospective study from Yoon et al. highlighted the effectiveness of targeting CSCs as a diagnostic/therapeutic approach in RC patients receiving preoperative CRT. Results suggested a correlation between the positivity to two surrogate markers (ALDH1 and CD44) and the regression grade of RC. Positivity to ALDH1 was also associated with short recurrence-free survival (RFS) and RC specific survival (RCS) [17]. ALDH1 expression levels were significantly higher in advanced RCs (stage III/IV) compared with early (stage II) tumors, as demonstrated in a more recent study from Vermani et al. focused on the identification/validation of suitable housekeeping genes for the evaluation of gene expression in RC [44]. In 2017, Luo et al. analysed primary tumours from 39 RC patients who received CRT and evaluated the in vitro stemness ability and the in vivo tumorigenic properties of sphere cells derived from the established colon cancer cell lines HT-29, HCT-116 and HCT-15. Study results showed that cells surviving to radiation treatment displayed high levels of G9a, a lysine methyltransferase involved in histone methylation, whose expression is positively correlated with CD133 in LARC patients. Knockdown of G9a increased the sensitivity of cells to radiation treatment, thus acting as a predictor of response to preoperative CRT in patients with advanced CRC [40]. Ganesh et al. investigated the radiosensitivity of organoid cultures (tumoroids) isolated and propagated from patients with primary, metastatic or recurrent RCs. Ex vivo responses to chemotherapy and radiotherapy correlated with the clinical responses observed in individual patients’ tumours. Furthermore, RC tumoroids recapitulated the heterogeneous response to radiotherapy observed in clinical settings [45]. 

The effect of radiation therapy on CRC stem cells (CR-CSCs) was published in two 2019 studies [32,46]. The first study from Chen et al. investigated the radiosensitising effect of polydatin (PD) by inducing apoptosis of CR-CSCs. The investigation was carried out on a C57BL/6 mouse model of CRC induced with azoxymethane/dextran sodium sulfate (AOM/DSS), which was divided into four groups: (i) control, (ii) PD only, (iii) IR only, (iv) PD + IR combination. Radiotherapy (dose rate: 2 Gy/min, 10 Gy) was administered once a week for a total of four times. In order to determine the radiosensitising mechanism of PD, CSCs were treated with the type I bone morphogenetic protein (BMP) receptor inhibitor K02288. Proliferation of Lgr5^+^ CSCs was dramatically increased by the addition of K02288. Both in vivo and in vitro experiments demonstrated that the combined IR + PD treatment inhibited the proliferation and promoted apoptosis of Lgr5^+^ CR-CSCs, suggesting an involvement of the BMP signalling pathway in the radiosensitising effect of PD [46]. In the second study, Anuja et al. assessed the radioresistance in CR-CSCs by irradiating parental HCT116 and HCT-15 cells and derived clonospheres and evaluated the DNA damage response (DDR) by γH2AX foci formation, COMET assay and ATM, p-ATM, and ERCC1. A higher survival rate in clonospheres post-irradiation compared to their parental counterpart, which corresponds to efficient DDR, was found. Differential expression of developmental markers, CSC markers (CD44, KLF4) and telomeric components were observed after irradiation, suggesting that they may contribute to the radioresistance property of CSCs [32]. In 2020, Endo et al. used a radiosensitivity assay in a study aimed to investigate the effect of radiation on growth, stemness and differentiation of four cancer tissue-originated spheroid (CTOS) lines (C45, CB3, C138, C111). As a consequence of a 9 Gy sublethal dose, a drastic reduction in proliferation and stemness markers (including Wnt target genes) and a persistence of differentiation markers were found. After a static growth phase, regrowth foci appeared, consisting of highly proliferating cells expressing stem cell markers (CD44v9 and Lgr5), which were the same that showed activated Wnt signalling at the time of irradiation [47]. Analysis of the stemness plasticity role in CTOSs radiosensitivity indicated a higher regrowth in disrupted compared to non-disrupted cell lines, thus suggesting a link between mechanical disruption and increased CTOSs’ radioresistance [47]. Moreover, pre-treating organoids with histone deacetylase inhibitors increased organoid radiosensitivity and the relevant suppression of Wnt signal-related gene expression. Taken together, these results suggested that foci can rise from a small subset of cancer cells with high Wnt activity at the time of irradiation; furthermore, the increased sensitivity of CTOS following pre-treatment with a Wnt inhibitor might have been due to alterations in stemness/differentiation status, rather than alterations in the DNA damage response. Finally, both radiosensitivity and ability to form foci showed good correlation with in vivo radiation sensitivity [47].

Along this line, Puglisi et al. demonstrated in the same year that in vitro and in vivo models based on patient-derived CSCs were able to predict individual responses to radiotherapy in LARC [48]. CSCs were isolated from CRC biopsies and injected subcutaneously into the flanks of thymic immunocompromised mice. Hence, the resulting tumour mass explanted from the animal was processed for histologic and immunohistochemical analyses and subjected to in vitro irradiation with the same clinical protocol used for LARC patients (5 Gy/Day) and the effects of the dose rate in terms of cell growth arrest and apoptosis induction were investigated. These in vitro results demonstrated that radiosensitivity varies among CSCs, with no differences among the various dose–rate protocols tested. Notably, the specific CSCs’ in vitro and in vivo sensitivity values corresponded to patients’ responses to radiotherapy [48].

## 4. Discussion

To the best of the authors’ knowledge, this is the first systematic review to collect the most relevant studies focusing on the CSC mechanisms of resistance to radiotherapy in RC. Four hundred and sixty-four studies published between January 2009 to July 2021 were systematically reviewed and only 11 were selected, suggesting the urgent need for further studies including larger cohorts of patients. There is currently a lack of studies regarding somatic mutations specific for the CSC population in RC, which are responsible for radiotherapy resistance. However, mutations in PI3K/AKT/mTOR signalling pathway components and in DNA damage response (DDR) as well as in DNA mismatch repair (MMR) candidate genes have been identified as potential mechanisms of CRT resistance in the bulk population of RCs [49,50]. Preoperative RT, administered as a conventional fractionated RT (45 Gy to the pelvis, followed by 5.4 Gy in 3 fractions to the tumour) has been recognised to play a key role in the standard multidisciplinary treatment of RC, by reducing local recurrence and increasing survival [51,52,53]. However, tumour response to RT differs considerably among patients, with several tumour types, including RC [39]. A significant number of patients exhibiting a complete response to neoadjuvant therapy experience local regrowth or metastatic dissemination, which is strongly associated with the capability of CSCs to resist treatment and promote cancer progression [48]. Serious adverse events such as anorectal and genitourinary complications derived from RT, may negatively impact on patient’s quality of life, despite the severity of these effects being affected by individual susceptibility, radiation dosing and accuracy. Hence, a proper identification of the patients who are likely to benefit from neoadjuvant chemo-radiotherapy as well as the most suitable therapy (dose or protocol) for each individual is important for managing patients and supporting clinical strategies [48]. To date, there are no indicators able to distinguish RC resistance and neoadjuvant treatment [40]. Therefore, identifying specific biomarkers to predict patient response to RT can help in planning a strategy aimed at targeting not only the tumour bulk, but also the sensibility of CSCs to RT, in order to avoid therapeutic failure and unnecessary treatments [20,31,39,40]. This review analysed the most recent studies focused on the characteristics of CSCs and therapeutic sensitivity targeting CSC radiosensitivity/radioresistance, through the detection of RT-response predictive biomarkers. Experimental data have shown a great heterogeneity in tumour radiosensitivity. Radioresistance, either intrinsic or acquired (leading to the development of adaptive responses induced by the irradiation itself), represents a barrier to overcome in order to improve both prognosis and treatment efficacy. The emerging role of CSCs in tumour response to RT has promoted the investigation of the molecular mechanisms underlying radioresistance in these cells [39,41]. To date, studies on cellular radiosensitivity in vitro have identified several cell-surface biomarkers of CSCs, showing differences between marker-negative and positive cells, the latter generally being more radioresistant. The techniques developed have identified some markers (summarised in Table 2), such as CD133, CD44, ALDH1, Lgr5 and G9a, whose positivity has been correlated with greater resistance to RT and poorer outcomes [17,32,40,41,42,46,47,48]. Tumour radioresistance, with consequent regrowth and spread, was seen to be strongly associated with the DNA-repairing ability of CSC, which promotes angiogenesis by enhancing vascular endothelial growth factor (VEGF) expression, or with the acidic microenvironment around hypoxic cells, which contributes to metastasis [39,41,54]. Current evidence indicates that CD133 expression correlates with poor RT response, thus suggesting the prognostic importance of this marker in the clinical setting [39]. CD44 expression also significantly correlated with poor survival and resistance to ionizing radiation, therefore indicating that other markers may be potential candidates of CSC radiosensitivity [39,46]. Lgr5 is linked with stemness and renewal but not with tumour progression. Recently, it was established as a surface marker for colorectal CSCs, particularly when co-expressed with CD44 and EPCAM [55]. G9a is involved in the DNA damage response, leading to the malignant phenotype of CRC. Since a significantly higher level of G9a expression was observed after radiation, it could be used as a predictive biomarker of radiosensitivity. Furthermore, a positive correlation between G9a and CD133 was noted in patients with LARC receiving neoadjuvant chemoradiotherapy. The knockdown of G9a resulted in a cell population with high sensitivity to radiation treatment [56]. ALDH1 positivity in residual disease after CRT could be a robust and an independent predictor of recurrence in residual rectal cancer patients treated with preoperative CRT followed by curative surgery [17]. Clinical studies analysed in this review showed that some parameters correlate with the outcome of RT and may be considered as surrogate markers for predicting CSC radiosensitivity. There is a growing evidence of the importance of cancer 3D organoid culture or in vivo models for cancer biology, preserving many of the biological and histological features and properties of the human tumours from which they were derived. Based on this evidence, an assay for stemness markers aimed at evaluating the effects of RT treatment in vitro could be translated in clinical decisions for the management of RC patients, thus avoiding radiation toxicity to resistant patients and reducing the treatment costs [48,57,58]. A recent study demonstrated that the in vivo responses to radiotherapy were fully consistent with those obtained in vitro, indicating that the intrinsic radiosensitivity grade of CSC isolated from RC needle biopsies was also maintained in the animal model [48]. For this reason, refining organoid culture techniques becomes necessary to make such methods as useful tools for biological analysis, in order to deepen the knowledge of mechanisms underlying radioresistance, as well as for therapeutic studies. This in vitro radiotherapy response predictivity assay could support clinical decisions for the management of RC patients, thus avoiding radiation toxicity to resistant patients and reducing the treatment costs. Thus, it is crucial to further investigate the individual impact of the association of chemotherapy to RT, which today represents the standard neoadjuvant treatment for patients with CR. Although it is clear that long-course CRT following the removal of the entire mesorectum is the best option for LARC, less evident is the recommendation for early rectal cancer located in the lower rectum or for patients bearing high-risk for cancer progression. Moreover, the role of chemotherapy intensification in improving pathological complete remission or in reducing distant failure rates should be further investigated [59]. 

The major limitation of the studies herein discussed is the small number of patients and samples evaluated. The median number of the examined samples was 46, ranging from 2 to 145. Even though a significant correlation between CSC biomarker expression and patients outcome was observed among the majority of studies analyzed, major efforts are required to enroll more patients into studies and transfigure preclinical observations into clinical practice. Another major issue to face is the heterogeneity in the experimental approach used to study the CSC population and its response to radiotherapy and, also, in the methods to detect and quantify CSC biomarkers expression. The diverse sensibility of techniques and the arbitrary scoring system used could compromise reliable comparisons among studies and jeopardize the association between CSC biomarkers expression and CRT sensibility/resistance [60]. Altogether the above mentioned concerns mirror the complexity in establishing the importance and reliability of biomarkers in the management of RC patients.

However, the studies selected in this systematic review highlight the usefulness of CSC-associated biomarkers as reliable predictors of therapy response in LARC patients. Further studies in the field comprising larger cohorts of patients will pave the way for more tailored therapies that avoid unnecessary treatments and side effects. 

## 5. Conclusions

Elevated expression of CSC biomarkers such as CD133, CD44, ALDH1, G9a and Lgr5 have been correlated with radioresistance in RC and with poor outcome. Only 11 studies have been selected in this systematic review based on the inclusion criteria and the low risk of bias. Further studies are needed to investigate additional CSC biomarkers and their specific characteristics that are predictive of radioresistance in RC patients. Based on current evidence, an organoid platform along with a stemness marker assay evaluating the effects of RT treatment in vitro could be essential to deepen the knowledge of mechanisms underlying radioresistance and to avoid unnecessary radiation toxicity in resistant RC patients.

## Figures and Tables

**Figure 1 genes-12-01502-f001:**
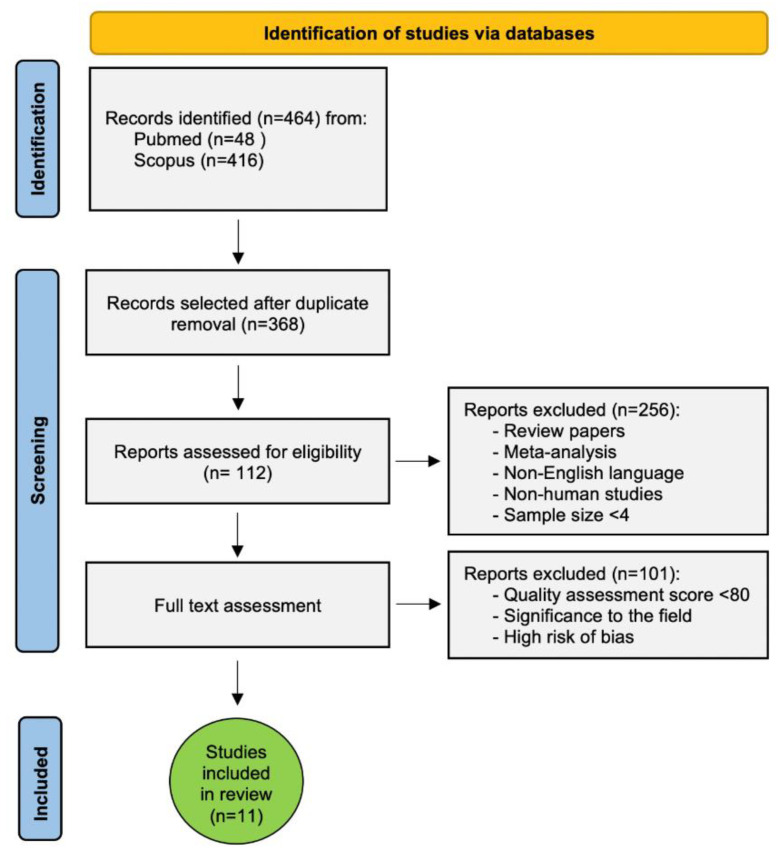
Flow chart of study selection strategy.

**Figure 2 genes-12-01502-f002:**
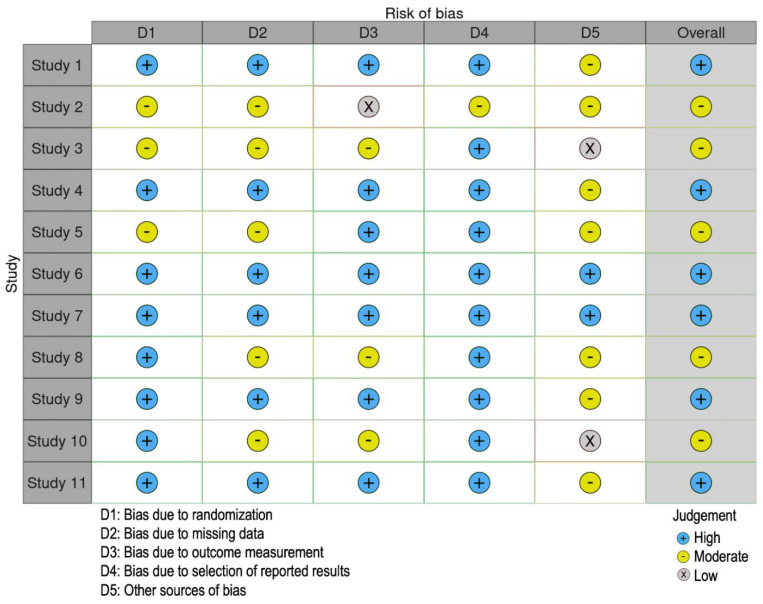
Risk of bias of the selected studies.

**Figure 3 genes-12-01502-f003:**
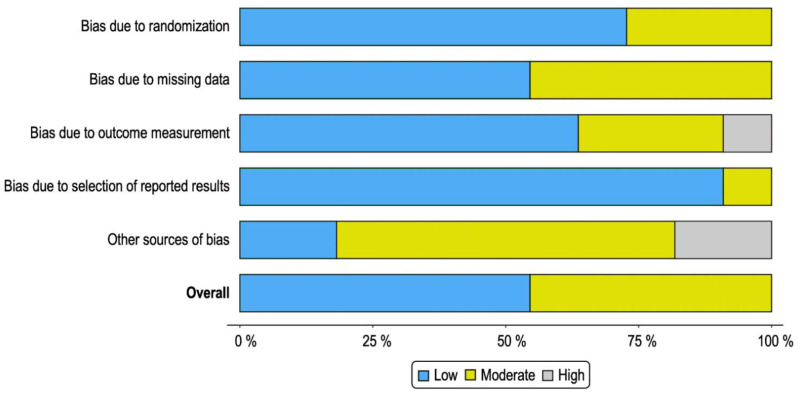
Summary plot of bias of the selected studies as a percentage of the total.

**Table 1 genes-12-01502-t001:** Criteria used to assess the quality and bias of the selected studies.

No.	Question	Answer
Q1	Is the study design well described?	Yes/No
Q2	Is the study well written and the English language of sufficient quality?	Yes/No
Q3	Is the experimental plan well organized?	Yes/No
Q4	Are the results statistically significant?	Yes/No
Q5	Are the positive/negative controls reported?	Yes/No
Q6	Do the findings support the conclusions of the study?	Yes/No
Q7	Are the human samples utilized ≥4?	Yes/No
Q8	Is the study significant for the field ?	Yes/No
Q9	Do the study cover the relevant literature in an unbiased manner?	Yes/No
Q10	Is there any other source of bias in the study?	Yes/No

**Table 2 genes-12-01502-t002:** Summary of the main in vitro/in vivo studies (2009–2021) investigating the radiosensitivity/radioresistance of CSCs in RC.

Study	Reference	Title of the Study	Patients and Samples	Biomarker	Biomarker Detection	Radiotherapy	Results
1	Saigusa S, Tanaka K, Toiyama Y, et al. 2009 [31]	Correlation of CD133, OCT4, and SOX2 in Rectal Cancer and Their Association with Distant Recurrence After chemoradiotherapy.	RC cells isolated from patients (TNM clinical stage II/III) pre- and postoperative CRT (*n* = 33).	CD133 OCT4 SOX2	IHC Real-time PCR	Dose rate: Preoperative radiotherapy at 20–45 Gy. Postoperative radiotherapy with short-course radiation in 28 patients (20 Gy, five fractions over a week) or fractionated radiation in 5 patients (45 Gy, 18 fractions for 4 weeks).	Significant positive correlation between post-CRT levels of CD133, OCT4 and SOX2 and disease-free survival probability (*p* = 0.0285; *p* = 0.0114; *p* = 0.006).
2	Chen T, Zhang Y, Guo WH, et al. 2010 [32]	Effects of heterochromatin in colorectal cancer stem cells on radiosensitivity.	Human colorectal adenocarcinoma samples from patients (*n* = 16).	CD133	Flow cytometry Immunofluorescence	Dose rate: 2 Gy/min (one side of the flank in nude mice was exposed to 10 Gy single dose of radiation, the other side without treatment served as control).	CSCs play a role in radiosensitivity in CRC, with a mechanism related to heterochromatin formation and histone methylation
3	Saigusa S, Tanaka K, Toiyama Y, et al. 2010 [33]	Immunohistochemical features of CD133 expression: Association with resistance to chemoradiotherapy in rectal cancer.	CSCs from RC patients (*n* = 50) and primary CRC patients (*n* = 40).	CD133	IHC	Dose rate: 1, 2.5, and 5 Gy.	Correlation between CD133 expression and histopathological response to preoperative CRT. CD133 was also associated with resistance to CRT.
4	Saigusa S, Inoue Y, Tanaka K, et al. 2012 [34]	Clinical significance of LGR5 and CD44 expression in locally advanced rectal cancer after preoperative chemoradiotherapy.	RC specimens obtained from patients who underwent preoperative CRT (*n* = 52).	LGR5 CD44	IHC	Dose rate: short-course (20 Gy in 4 fractions) or long-course (45 Gy in 25 fractions) radiotherapy.	Gene expression levels of LGR5 in cancer cells and CD44 in cancer stroma were significantly correlated with disease recurrence. High expression levels of stromal CD44 was an independent prognostic factor of recurrence and overall survival of RC patients after preoperative CRT.
5	Yoon G, Kim SM, Kim HJ, et al. 2016 [17]	Clinical influence of cancer stem cells on residual disease after preoperative chemoradiotherapy for rectal cancer.	Surgical specimens from patients with residual RC after CRT (*n* = 145).	ALDH1 CD44	IHC	Dose rate: long-course radiation, 45 or 50 Gy in 25 fractions of 1.8 or 2 Gy administered to the whole pelvis five times per week for 5 weeks.	ALDH1 and CD44 positivity was related to lower TRG (*p* = 0.009; *p* = 0.003). ALDH1 positivity was associated short RFS and RCSS (*p* = 0.005 and 0.043 vs. *p* = 0.725 and 0.280, respectively). ALDH1 positivity was an independent prognostic factor for inferior RFS but not RCSS ((*p* = 0.039 vs. *p* = 0.571 [HR, 2.997; 95% CI, 1.059–8.478]).
6	Luo CW, Wang JY, Hung WC, et al. 2017 [35]	G9a governs colon cancer stem cell phenotype and chemoradioresistance through PP2A-RPA axis-mediated DNA damage response.	Primary tumors from patients who received preoperative CRT (*n* = 39) for RC and colorectal cancer cell lines (*n* = 3).	G9a CD133	IHC Real-time PCR Flow cytometry	Dose rate: 20–45 Gy pelvic RT	Significantly positive correlation between G9a and CD133 in locally advanced RC patients receiving preoperative CRT. Knockdown of G9a increased the radiosensitivity of cells and sensitised cells to DNA damage agents through PP2A-RPA axis.
7	Ganesh K, Wu C, O’Rourke KP, et al. 2019 [36]	A rectal cancer organoid platform to study individual responses to chemoradiation.	RC tumoroids (*n* = 65) from *n* = 41 patients (22 from treatment-naïve patients; 43 from patients undergoing first- or second-line therapy) and normal rectal organoids from normal adjacent tissue (*n* = 51)	CDX2, nuclear β-catenin, Alcian blue, MUC-2, CK20, E-cadherin	IHC Immunofluorescence	Dose rate: 250 kVp and 12 mA	RC tumoroids display varying sensitivity to ionizing radiation, which corresponds to clinical radiotherapy responses.
8	Chen Q, Zeng YN, Zhang K, et al. 2019 [37]	Polydatin Increases Radiosensitivity by Inducing Apoptosis of Stem Cells in colorectal cancer.	C57BL/6 mouse model of CRC induced with AOM/DSS; CT26 and HCT116 colon cancer cells (*n*= 2).	Lgr5	Flow cytometry	Dose rate: 10 Gy, 2 Gy/min, once a week for a total of four times	IR plus polydatin inhibit the proliferation and promote apoptosis of Lgr5^+^ CR-CSCs through the BMP signalling pathway
9	AnujaK, Chowdhury AR, Saha A, et al.2019 [38]	Radiation induced DNA damage response and resistance in colorectal cancer stem-like cells.	HCT116 and HCT-15 cells and derived clonospheres (*n* = 2).	CD44 KLF4 β-catenin TRF2 RAP1 hTERT	Real-time PCR Immunofluorescence	Dose rate: 4.0 Gy/min ([0–8 Gy] of 6Mv energy X-rays)	CSCs endowed with high DNA repair capacity survive following radiation therapy
10	Endo H, Kondo J, Onuma K, et al. 2020 [39]	Small subset of Wnt-activated cells is an initiator of regrowth in colorectal cancer organoids after irradiation.	Cancer tissue originated spheroid derived from CRC specimens (*n* = 4).	CD44v9, Wnt target genes, Lgr5	Immunohistochemistry Immunofluorescence RT-PCR Real-time PCR	Dose rate: 9 Gy	Radiosensitivity differed among CTOS lines and showed good correlation with in vivo radiation sensitivity. Pre-treating organoids with HDACi increased radiosensitivity. Wnt inhibitors increased organoid radiosensitivity.
11	Puglisi C, Giuffrida R, Borzì G, et al. 2020 [40]	Radiosensitivity of cancer stem cells has potential predictive value for individual responses to radiotherapy in locally advanced rectal cancer.	CSC lines (*n* = 4) from CRC biopsies; animal models, generated by CSC xenotransplantation.	CD44, CD133	Flow cytometry	Dose rate: Fractioned 25 Gy dose administered daily (5 Gy/Day) CSCs and animal models were subjected to in vitro irradiation with the same clinical protocol used for LARC patients	In vitro CSC radiosensitivity correspond to radiosensitive tumour xenografts upon subcutaneous implantation. CSCs’ in vitro and in vivo sensitivity values correspond to patients’ responses to radiotherapy.

Abbreviations: CRT chemoradiotherapy; RT rectal cancer; AOM/DSS, azoxymethane/dextran sodium sulfate; CI, confidence interval; CRC, colorectal cancer; CSC, cancer stem cells; CR-CSCs, colorectal cancer stem cells; CTOS, cancer tissue-originated spheroid; HCT, human colorectal carcinoma cell line; HDACi histone deacetylase inhibitors; HR, hazard ratio; IHC, immunohistochemistry; LARC, locally advanced rectal cancer; RC, rectal cancer; RCSS, rectal cancer specific survival; RFS, recurrence-free survival; RT, radiotherapy; RT-PCR, Reverse transcriptase PCR; TRG, tumour regression grade.

## Data Availability

Not applicable.

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
