# Peer review of "Cancer Stem Cell Biomarkers Predictive of Radiotherapy Response in Rectal Cancer: A Systematic Review"

_genes, 2021, doi:10.3390/genes12101502_

Round 1

Reviewer 1 Report

This manuscript covers the important recent studies in the subject matter, and thoroughly and comprehensively covers the topic. The list of references is comprehensive and the material is appropriately referenced. Just minor changes are required as listed below.

Line 22: Consider changing “despite” to “although” to allow flow in the sentence.

Line 31: Consider adding “provide an” before overview and “of” after overview for better flow.

Line 49: Consider removing “Despite”

Line 53: “Moreover, a limited number of patients respond positively….”

Lines 57-59: Consider rephrasing

Line 93: Consider substituting “Thank” with “Due”

Lines 96 to 97: Consider rephrasing

Line 210: “...identification/validation of suitable…”

Line 288: “…neoadjuvant therapy, experience ...”

Lines 290 – 293: “…these effects being affected…”

Line 314: “…which contributes to…”

Author Response

This manuscript covers the important recent studies in the subject matter, and thoroughly and comprehensively covers the topic. The list of references is comprehensive and the material is appropriately referenced. Just minor changes are required as listed below.

Line 22: Consider changing “despite” to “although” to allow flow in the sentence.

Line 31: Consider adding “provide an” before overview and “of” after overview for better flow.

Line 49: Consider removing “Despite”

Line 53: “Moreover, a limited number of patients respond positively….”

Lines 57-59: Consider rephrasing

Line 93: Consider substituting “Thank” with “Due”

Lines 96 to 97: Consider rephrasing

Line 210: “...identification/validation of suitable…”

Line 288: “…neoadjuvant therapy, experience ...”

Lines 290 – 293: “…these effects being affected…”

Line 314: “…which contributes to…”

We thank Reviewer #1 for the constructive comments.

All the suggested changes have been made to the revised version of the manuscript.

Reviewer 2 Report

Are there any other markers of CRCs which are related to radioresistance, if yes, why authors chose not to include them?

Author Response

Are there any other markers of CRCs which are related to radioresistance, if yes, why authors chose not to include them?

We thank the Reviewer #2 for the constructive comments. The aim of this systematic review was to overview the current scenario of in vitro and in vivo studies that analyze the biomarkers predictive of RT-response in cancer stem cells derived from RC patients. Considering that studies involving cancer stem cells are limited, we selected 11 recent studies by following the strict criteria reported in the material and methods section of our review article. Other markers of CRCs related to radioresistance, which are not specific for stem cells and thus out of scope of this review article, have been described (doi: 10.1016/j.ijrobp.2009.03.003).

Reviewer 3 Report

In this review article, the authors investigated and overviewed the most recent studies focusing on the cancer stem cells mechanisms of resistance to radiotherapy in rectal cancer. The prognostic value of cancer stem cells (CSC) biomarkers as CD133, CD44, ALDH1, Lgr5 and G9a have been previously reported in the last decade to associate with survival in many cancer types, such as patients with colorectal cancer. Here, the authors sought to highlight the clinical significance of those CSC biomarkers in rectal cancer, and conclude that further studies including larger cohort of patients are needed to investigate more CSC biomarkers and evaluate the biomarkers predictive of radiotherapy response in cancer stem cells.

Minor issues:

Page 8, line 183: the second part of the sentence " showing differences between marker-negative and positive cells, the latter generally being more radioresistant " is vague and can be moved to and explained in the discussion section. 

Author Response

In this review article, the authors investigated and overviewed the most recent studies focusing on the cancer stem cells mechanisms of resistance to radiotherapy in rectal cancer. The prognostic value of cancer stem cells (CSC) biomarkers as CD133, CD44, ALDH1, Lgr5 and G9a have been previously reported in the last decade to associate with survival in many cancer types, such as patients with colorectal cancer. Here, the authors sought to highlight the clinical significance of those CSC biomarkers in rectal cancer, and conclude that further studies including larger cohort of patients are needed to investigate more CSC biomarkers and evaluate the biomarkers predictive of radiotherapy response in cancer stem cells.

Minor issues:

Page 8, line 183: the second part of the sentence " showing differences between marker-negative and positive cells, the latter generally being more radioresistant " is vague and can be moved to and explained in the discussion section.                          

We thank Reviewer #3 for the valuable comment.

The indicated sentence has been moved and further explained in the discussion section.

Reviewer 4 Report

Thank you for the manuscript!

The bias discussed by yourself when comparing different investigation methods and randomization procedures, cannot be neglected. 
The small number of patients studied and the different experimental approach basically do not allow a valid conclusion.
Furthermore, different dose and fractionation schemes make it difficult to draw conclusions about the significance of the biomarkers. As much as they try to be correct in their technical evaluation, considering the complexity of the question, many questions remain unanswered.

- Although it is presented how useful biomarkers would be in the treatment of patients with rectal cancer, the discussion should explain the complexity of the question in much more detail. 
- In addition, the different measurement methods should be elaborated and compared.
- The influence of chemotherapy usually combined with radiotherapy on outcomes is hardly described or discussed. This should be done in much more detail. 
- Table 2: Instead of the column with the study title, the detection procedure of the respective study should be described. A clearer identification of the number of cases n would also be welcome.
- As a sufferer of deuteranopia, I plead for a different coloring in figures 2 and 3.

Author Response

Thank you for the manuscript!

The bias discussed by yourself when comparing different investigation methods and randomization procedures, cannot be neglected. The small number of patients studied and the different experimental approach basically do not allow a valid conclusion.
Furthermore, different dose and fractionation schemes make it difficult to draw conclusions about the significance of the biomarkers. As much as they try to be correct in their technical evaluation, considering the complexity of the question, many questions remain unanswered.

We thank this Reviewer for the productive comments.

- Although it is presented how useful biomarkers would be in the treatment of patients with rectal cancer, the discussion should explain the complexity of the question in much more detail.

The complexity in summarizing the role of cancer stem cell biomarkers expression in the prediction of chemoradiotherapy response in rectal cancer has been addressed in the Discussion section of the revised version of this manuscript. 

- The influence of chemotherapy usually combined with radiotherapy on outcomes is hardly described or discussed. This should be done in much more detail.

The role of chemotherapy in combination with radiotherapy in influencing the outcome of rectal cancer patients, with different prognosis, has been further examined in the discussion section of this manuscript.

- In addition, the different measurement methods should be elaborated and compared.
Table 2: Instead of the column with the study title, the detection procedure of the respective study should be described. A clearer identification of the number of cases n would also be welcome.

The biomarker measurement methods used in each study have been specified in the new Table 2 and the challenge in using different experimental and methods to detect and quantify CSCs biomarkers expression has been addressed in the Discussion section. In table 2, the number of cases for each study have been indicated.

- As a sufferer of deuteranopia, I plead for a different coloring in figures 2 and 3.

As suggested by this reviewer, Figure 2 and 3 have been newly colored.

Round 2

Reviewer 4 Report

The authors have made all the changes I suggested. I support the publication of the manuscript in its present form.